# Assessment of Microplastics in a Municipal Wastewater Treatment Plant with Tertiary Treatment: Removal Efficiencies and Loading per Day into the Environment

**Javier Bayo \*** 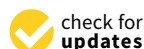**, Sonia Olmos and Joaquín López-Castellanos**

Department of Chemical and Environmental Engineering, Technical University of Cartagena,
Paseo Alfonso XIII 44, E-30203 Cartagena, Spain; soniaespinar19@gmail.com (S.O.); qlopezca@gmail.com (J.L.-C.)
\* Correspondence: javier.bayo@upct.es

**Abstract:** This study investigates the removal of microplastics from wastewater in an urban wastewater treatment plant located in Southeast Spain, including an oxidation ditch, rapid sand filtration, and ultraviolet disinfection. A total of 146.73 L of wastewater samples from influent and effluent were processed, following a density separation methodology, visual classification under a stereomicroscope, and FTIR analysis for polymer identification. Microplastics proved to be 72.41% of total microparticles collected, with a global removal rate of 64.26% after the tertiary treatment and within the average retention for European WWTPs. Three different shapes were identified: i.e., microfiber (79.65%), film (11.26%), and fragment (9.09%), without the identification of microbeads despite the proximity to a plastic compounding factory. Fibers were less efficiently removed (56.16%) than particulate microplastics (90.03%), suggesting that tertiary treatments clearly discriminate between forms, and reporting a daily emission of $1.6 \times 10^7$ microplastics to the environment. Year variability in microplastic burden was cushioned at the effluent, reporting a stable performance of the sewage plant. Eight different polymer families were identified, LDPE film being the most abundant form, with 10 different colors and sizes mainly between 1–2 mm. Future efforts should be dedicated to source control, plastic waste management, improvement of legislation, and specific microplastic-targeted treatment units, especially for microfiber removal.

**Keywords:** microplastic; wastewater; tertiary treatment; microfiber; oxidation ditch; RSF; UV disinfection

## 1. Introduction

Plastic pollution is a widespread problem, mainly affecting the oceans but also human health, food safety, and climate change, and its world production has been reported to have grown from 359 million tons in 2018 to 368 million tons in 2019 [1]. Microplastics (MPs), first identified by [2] in the Sargasso Sea and defined as particles with a size below 5 mm in their longest dimension [3], can originate from the decomposition of macroplastics (secondary MPs) or be intentionally manufactured by different firms with a microscopic size (primary MPs) [4]. Primary MPs are used in the form of spherules as precursors in the plastic industry, with a wide range of applications such as packaging, office equipment, and vehicle construction, or as scrubbing material in personal care products or air-blasting granules and pellets, among other uses [4]. Secondary MPs are formed by means of chemical and physical mechanisms such as hydrolytic degradation, photolysis, weathering, ultraviolet radiation, or abrasion [5,6], or via biotic degradation, as biodeterioration [7,8]. Microplastics are considered to be more prevalent in the environment than macro or mesoplastics, owning their larger quantities and small sizes [9].

The European plastic industry, including plastics raw materials, producers, converters, recyclers, and machinery manufacturers, gives direct employment to more than 1.56 million people, and plastic production and use have an important role in a more sustainable future because of the unique properties of plastics [1]. At the same time, its commitment

as an industry is to avoid plastic waste, having taken important steps to understand the true nature of durable and degradable plastic materials and their behaviors in the environment [10].

Microplastic pollution is a topical issue of global concern both in marine and freshwater ecosystems; i.e., rivers, beaches, reefs, lakes, surface water, estuaries, or lagoons [6,11–13]. Furthermore, the sorption of toxic chemicals by MPs has been thoroughly reported, tending to be ingested by different organisms [14].

Wastewater treatment plants (WWTPs) act as a sink for MPs from both domestic and industrial wastewater, but also as a source of MPs for the environment and freshwater [3,15–17], proven by the large quantities of microfibers and secondary MPs reported close to wastewater effluents and mainly originating from laundry [18]. Several studies have shown an effective MP elimination from effluent, with a great variation of removal percentages; i.e., 53.6% [19], 57% [20], 64.4% [21], 72% [22], 90.3% [17], 95.16% [23], or 99.9% [24], among many others.

Sewage plants are complex systems, with chemical, physical, and biological processes taking place simultaneously, and the removal efficiency relies on different treatments, including skimming and settling processes, primary clarification, biological removal, and tertiary treatments [16,24,25]. However, synthetic microfibers are virtually everywhere in the environment [18], and the ability of tertiary treatments to efficiently and significantly remove these microfibers is yet to be assessed. Domestic washing of textiles and garments is a constant and widespread source of plastic microfibers [18] released from washing machines to wastewater treatment plants, especially during Autumn and Winter [26,27] and it is estimated that over 1900 fibers are released per washing cycle and garment [26], varying from 120 to 728,289 particles from similar garments [28].

The aim of this study was to monitor a full-scale WWTP in Southeast Spain, providing basic information on the abundance, shape, size, color, and type of MPs in influent (INF) and effluent (EFF) samples, after an oxidation ditch system, rapid sand filtration (RSF), and ultraviolet disinfection. We assessed the commitment of the plant to European MP standards in wastewater systems, the importance of tertiary treatments in microfibers (FB) release, and the influence of the surrounding environment in close proximity to the sewage plant; i.e., a plastic compounding factory, in an expanding urban area with greenhouse agricultural crops, the correspondence between demanded and collected polymer types, and their possible sources. In addition, some solutions are proposed in order to improve the removal efficiency of these emerging pollutants.

## 2. Materials and Methods

### 2.1. Description of "La Aljorra" WWTP and Sampling Collection

"La Aljorra" is a full-scale WWTP treating both domestic and industrial wastewater, located in the Region of Murcia (Southeast Spain) (37°41′16″ N, 01°03′13″ W) (Figure S1). It is designed with a maximum hydraulic flow of 677 m$^3$ h$^{-1}$ and serving about 70,417 equivalent inhabitants. Pretreatment includes bar screens for both rough and fine solids, and grit and grease removal with aeration supplied by blowers. After pretreatment, wastewater is introduced in parallel into a two full-scale oxidation ditch system (8502 m$^3$ each), equipped with an internal pre-anoxic zone. The mixed liquor leaves the oxidation ditch to secondary settlers, and clarified effluent ends up in rapid sand filters with a total filtering surface of 320 m$^2$. For a detailed description of the sewage treatment line in "La Aljorra" WWTP and physicochemical and biological parameters, please refer to the Supplementary Material (Tables S1 and S2).

A total of 146.73 L of wastewater were collected for the study through 28 grab samples processed between 28 February 2019 and 20 May 2020, and distributed into 14 samples from influent (INF = 59.89 L) and 14 samples from effluent (EFF = 86.84 L) (Table S3). Sample volumes were accurately measured for each experiment, ranging from 2.40 to 6.23 L for INF (mean ± standard error) (4.28 ± 0.23 L), and 3.00 to 8.89 L for EFF (6.20 ± 0.42 L), always collected on Thursday morning (9–11 a.m.) in glass bottles with a metallic lid.

Samples from EFF were directly vacuum filtered through a Büchner funnel using a 0.45 μm paper filter (Prat Dumas, Couze-St-Front, France, 110 mm Ø). INF samples were previously treated with an environmentally-friendly, cheap, and inert salt-saturated solution 120 g L$^{-1}$ NaCl (2.05 M) (1.08 g cm$^{-3}$) (Panreac, Barcelona, Spain), in a methodology wholly reported in [17] and detailed in Figure 1. All experiments were carried out at room temperature (293 K).

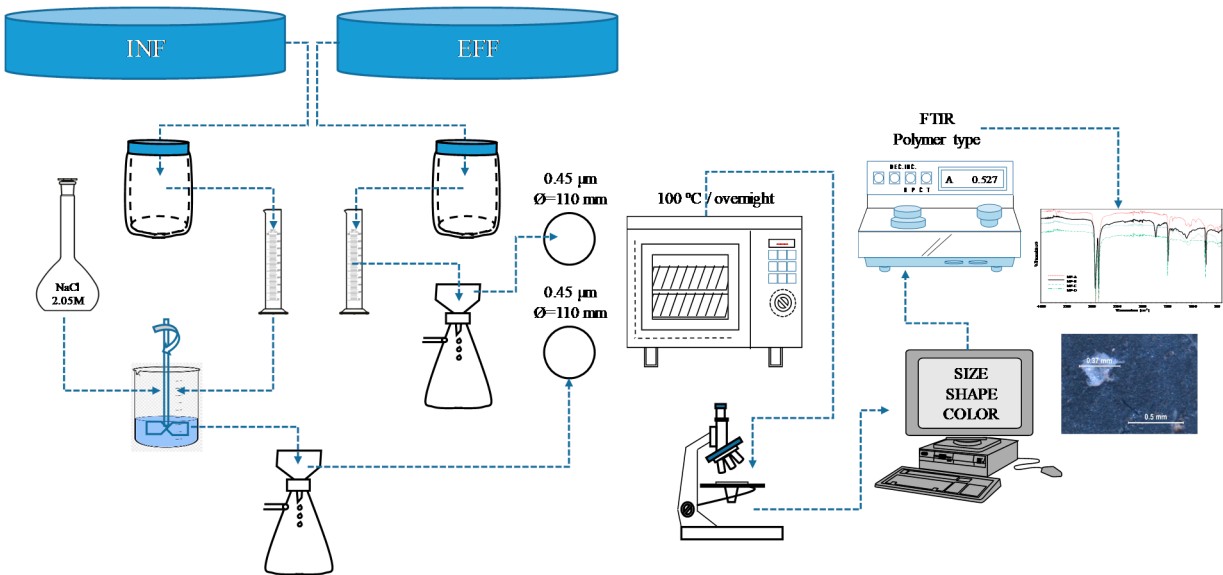

**Figure 1.** Flow diagram for the analysis of microplastics in "La Aljorra" WWTP.

In order to prevent any risk of contamination, especially to minimize exposure to airborne MP, all equipment was covered with aluminum foil and filtered samples were placed into covered glass Petri dishes for further examination. All glassware was thoroughly washed with tap water and twice with deionized water after each experiment, and during the whole process, analysts only wore natural fabric clothes, nitrile gloves, clean cotton lab-gown, and face masks covering the nose and mouth. Procedural blanks, consisting of 1.5 L of 0.45 μm filtered deionized water, were processed in parallel with pair of wastewater samples (influent and effluent), and placed into clean glass Petri dishes for further microscopic examination. Only one fiber was quantified and subtracted from the results.

*2.2. Microplastic Analysis and Dataset*

Samples with possible MP were examined under a digital optical trinocular microscope (Olympus SZ-61TR Zoom, Olympus Co., Tokyo, Japan) coupled to a Leica MC190 HD digital camera and an image capturing software Leica Application Suite (LAS) 4.8.0 (Leica Microsystems Ltd., Heerbrugg, Switzerland), used for the analysis and recording of color, shape, and size of microparticles in their largest dimension. The analyzed microlitter (ML) was isolated in 40-mm glass Petri dishes for further study by FTIR. Microparticles were visually classified as microfiber (FB), microbead or pellet (BD), fragment (FR), and sheet or film (FI), allowing a maximum length of 15 mm for fibers, as proposed by [29].

FTIR was used for the identification of functional groups and molecular composition of polymeric surfaces. Samples were compressed in a diamond anvil compression cell, and spectra were acquired with a Thermo Nicolet 5700 Fourier transformed infrared spectrometer (Thermo Nicolet Analytical Instruments, Madison, WI, USA), provided with a deuterated triglycine sulfate, DTGS, detector, and KBr optics. The spectra collected were an average of 20 scans with a resolution of 16 cm$^{-1}$ in the range of 400–4000 cm$^{-1}$. Spectra were controlled and evaluated by the OMNIC software without further manipulations, and polymers were identified by means of different reference polymer libraries, as further

explained. All statistical analyses were carried out with SPSS 26.0 statistic software (IBM Co. Ltd., North Castle, NY, USA), with a critical value for statistical significance set at $p < 0.050$.

## 3. Results and Discussion

### 3.1. General Considerations and Removal Rates According to Major Shapes

A total of 319 microlitter particles (ML) were isolated from all wastewater samples, with an average concentration of $2.58 \pm 0.40$ items $L^{-1}$ and minimum and maximum values corresponding to 1.33 and 6.37 items $L^{-1}$ for INF, and 0.10 and 7.00 items $L^{-1}$ for EFF, respectively. Average concentrations were $3.78 \pm 0.48$ items $L^{-1}$ for INF (219 items) and $1.38 \pm 0.48$ items $L^{-1}$ for EFF (100 items), with a statistically significant removal rate of 63.41% (*F-test* = 12.509, $p = 0.002$). Every single isolated microlitter particle was analyzed by FTIR, and 72.41% of these 319 microparticles ($n = 231$) were identified as MP, with an average concentration of $1.86 \pm 0.32$ items $L^{-1}$. Among non-polymeric microparticles, chipboard and glass fragments, silica, cellulose from toilet papers and calcium stearate, stearic acid, and surfactants [30] from soap residues, cosmetics, and personal care products were mainly identified. Furthermore, Zn/Ca PVC stabilizer was also identified as an additive in the particulate fraction. It is a heat stabilizer, incorporated during PVC synthesis in order to rein back the polymer degradation during thermal processing [31]. Additives within the polymer matrix may leach into the environment during the lifetime of the product, causing alterations in freshwater microorganisms [32]. Figure 2 shows ML and MP images from samples collected at "La Aljorra" WWTP, proving both are optically similar and an additional spectroscopic technique is necessary to further minimize overestimation of suspected microplastics.

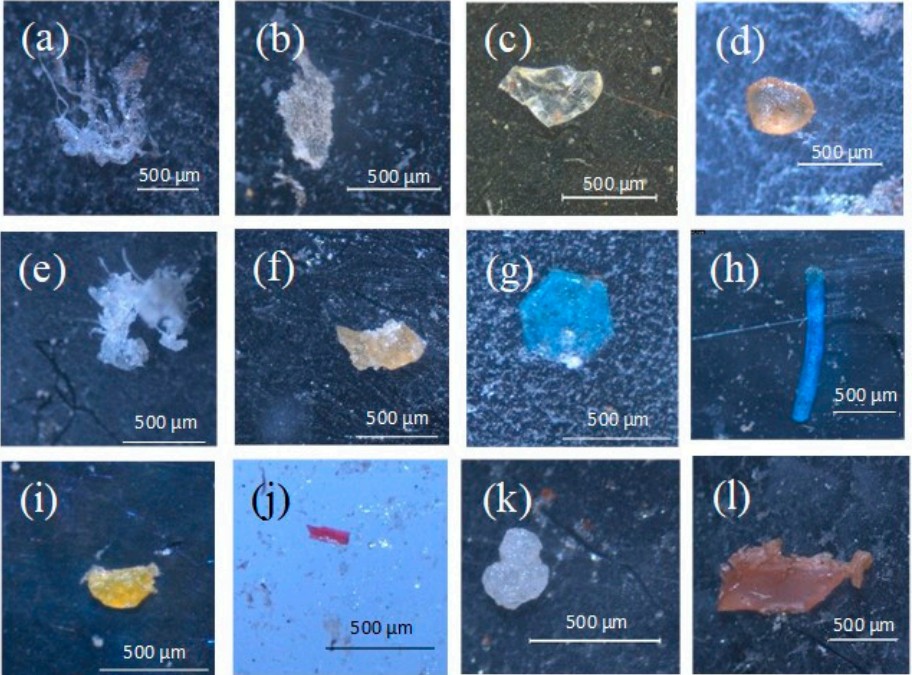

**Figure 2.** Microlitter (**a**–**d**) and microplastics (**e**–**l**) in "La Aljorra" WWTP: (**a**) cellulose (INF/1 August 2019); (**b**) chipboard (INF/20 February 2020); (**c**) silicic acid, sodium salt (EFF/20 February 2020); (**d**) calcium stearate (INF/19 December 2019); (**e**) poly(butyl acrylate (INF/12 March 2020); (**f**) sebacic acid biopolymer (INF/20 March 2020); (**g**) polyester (INF/19 September 2019); (**h**) polyethylene filament (EFF/17 October 2019); (**i**) polyester (EFF/19 September 2019); (**j**) polypropylene (INF/1 August 2019); (**k**) polyethylene wax (INF/20 February 2020); (**l**) oxidized polyethylene (INF/20 February 2020).

These results reinforce the need for additional spectroscopic techniques to objectively differentiate microplastics from non-plastic microparticles in the whole sample; i.e., FTIR or Raman spectroscopy [25]. Table 1 shows the microplastic content in both INF and EFF of "La Aljorra" WWTP. Statistically significant differences were observed between average MP concentration collected from INF ($2.74 \pm 0.49$ items $L^{-1}$) and those collected in the EFF ($0.98 \pm 0.27$ items $L^{-1}$) (*F-test* = 9.998, *p* = 0.004), indicating a removal rate of 64.26% for all microplastic forms, within the average retention rate in Europe for WWTPs of between 53% and 84% [32].

**Table 1.** Total count, percentages, and average concentrations ($\pm$ standard error) of microplastics in the influent (INF) and effluent (EFF) of "La Aljorra" WWTP. The concentrations are presented in items per liter of wastewater.

| | INF | EFF | TOTAL |
|---|---|---|---|
| Microplastic (MP) | 161 (69.70%) | 70 (30.30%) | 231 |
| | 2.74 ($\pm$0.49) | 0.98 ($\pm$0.27) | 1.86 ($\pm$0.32) |
| Fibes (FB) | 119 (64.67%) | 65 (35.33%) | 184 |
| | 2.09 ($\pm$0.47) | 0.92 ($\pm$0.26) | 1.50 ($\pm$0.29) |
| Microplastic particles (MPP) | 42 (89.36%) | 5 (10.64%) | 47 |
| | 0.66 ($\pm$0.14) | 0.07 ($\pm$0.03) | 0.36 ($\pm$0.09) |
| Film (FI) | 21 (80.77%) | 5 (19.23%) | 26 |
| | 0.35 ($\pm$0.10) | 0.07 ($\pm$0.03) | 0.21 ($\pm$0.06) |
| Fragment (FR) | 21 (100%) | 0 (0%) | 21 |
| | 0.31 ($\pm$0.09) | 0 | 0.15 ($\pm$0.05) |

In [33], a concentration of $1.0 \pm 0.4$ items $L^{-1}$ was reported in the final effluent of a Finnish WWTP with a daily average flow of 10,000 m$^3$, and [23] reported $0.59 \pm 0.22$ items $L^{-1}$ in the effluent from the largest water reclamation plant in China (1,000,000 m$^3$ day$^{-1}$). Minimum and maximum values corresponded to 0.65 and 6.37 items $L^{-1}$ for INF, and 0.17 and 3.33 items $L^{-1}$ for EFF, respectively. Concentrations in the EFF are consistent with those previously reported by our research group in another WWTP with RSF technology; i.e., between 0 and 5.28 items $L^{-1}$ [16]. Reported concentrations of MPs in final effluents from different WWTPs are highly variable according to the wastewater treatment technology applied, but also because of different sampling strategies or digestion procedures, among others. In [34], 97% MP removal by RSF is reported (from 0.7 to 0.02 items $L^{-1}$), [35] reports 73.8% removal using an Al-based coagulant, while [36] revealed a slight reduction after RSF treatment in an Israeli WWTP, with a final concentration of 1.9 items $L^{-1}$.

Three different shapes were isolated in wastewater samples: microfiber (FB) (79.65%; $1.50 \pm 0.29$ items $L^{-1}$), film (FI) (11.26%; $0.21 \pm 0.06$ items $L^{-1}$), and fragment (FR) (9.09%; $0.15 \pm 0.05$ items $L^{-1}$), similar to that described by [33] in a pilot-scale membrane bioreactor, [27] in a municipal WWTP in Turkey, or by [25] in 6 WWTPs in Guangzhou (China). Our results are also in agreement with those previously reported in another WWTP from our region, which also used rapid sand filtration as a tertiary treatment [16], but different from those reported in a conventional activated sludge sewage plant, where fragment form was mainly isolated [17]. In the present study, the only rounded or microbead form isolated was for a calcium stearate from soap residues, as shown in Figure 2d. This is consistent with the prohibition in most countries of personal care products with microbeads, in addition to a change in the trend and public awareness to use cleansing products with natural scrubbers [13,25]. Furthermore, preventive tools developed by the plastic factory close to the WWTP, such as the installation of filters at drains or immediately cleaning spilled pellets, have proven to be decisive and very efficient. These preventative tools have been adopted by firms that have signed the PlasticsEurope OCS (Operation Clean Sweep®) pledge since 2017, which is a voluntary industry initiative for preventing plastic pellet loss [37]. Regardless, there is no clear relationship between MP abundance in wastewaters and factories [25].

Particulate forms of microplastics (MPPs), including both fragments and films, had a statistically significant decrease from INF ($0.66 \pm 0.14$ items L$^{-1}$) to EFF ($0.07 \pm 0.03$ items L$^{-1}$) (*F-test* = 17.364, *p* = 0.000) (Table 1), accounting for a removal percentage of 90.03%. On the contrary, FB displayed a low removal efficiency of 56.16%, also significantly decreasing from INF ($2.09 \pm 0.47$ items L$^{-1}$) to EFF ($0.92 \pm 0.26$ items L$^{-1}$) (*F-test* = 4.798, *p* = 0.038) and confirming better settling rates for MPP than for FB. In fact, there was an increase in the abundance of microfibers in EFF (92.86%) compared to INF (73.91%), similar to that reported by [36] after a tertiary treatment (91%) relative to raw wastewater (74%). In [38], a higher prevalence of fibers than particles is also reported across three WWTPs in Charleston Harbor (USA). The average amount of microplastics released with the EFF was calculated as $1.6 \times 10^7$ MP per day, similar to other reported concentrations in different WWTPs; i.e., $6.5 \times 10^7$ MP per day [39], $3.6 \times 10^6$–$1 \times 10^7$ MP per day [5], $1 \times 10^7$ MP per day [33], $1.2 \times 10^7$ MP per day [40], or $4.15 \cdot 10^7$ MP per day [41].

Meanwhile, FR completely disappeared in the EFF of the WWTP, after the RSF and UV disinfection, some film forms, less dense than fragments, still appeared, decreasing from INF ($0.35 \pm 0.10$ items L$^{-1}$) to EFF ($0.07 \pm 0.03$ items L$^{-1}$) and showing a statistically significant removal rate of 81.35% (*F-test* = 7.722, *p* = 0.010). Similar results were reported by [27] for the overall plant operation, with removal rates decreasing from fragments to films and fibers.

Despite the number of microplastics trapped between the sand grains or adhered to their surfaces by interception, the main mechanism underlying rapid sand filtration [42] could be substantially enhanced with the use of coagulant [34,35]. Although "La Aljorra" WWTP is prepared for that procedure, no coagulant additions were used.

The global removal percentage for this study (64.26%) proved to be similar to that reported by [43] in two WWPTs located in Wuhan (China), 62.7% and 66.1%. The removal rate for FB (56.16%) was similar to that reported in our previous study (53.83%) [16], suggesting that tertiary treatments are, overall, more efficient than fiber forms in removing particulate [44]. In fact, while the average ratio MPP:MP decreased from INF ($0.28 \pm 0.05$) to EFF ($0.14 \pm 0.08$), the FB:MP relationship increased after the treatment process, from $0.72 \pm 0.05$ (INF) to $0.86 \pm 0.08$ (EFF).

Figure 3 shows the annual mean variability of MPP and FB vs. MP ratios for INF and EFF of "La Aljorra" WWTP. A statistically significant variation in both ratios could be observed for the influent, with an increase in MPP:MP ratio from 2019 ($0.21 \pm 0.05$) to 2020 ($0.45 \pm 0.07$) (*F-test* = 6.801, *p* = 0.023) and a decrease in FB:MP ratio from 2019 ($0.79 \pm 0.05$) to 2020 ($0.55 \pm 0.07$) (*F-test* = 6.804, *p* = 0.023).

This sewage plant is located in an expanding urban area, undergoing rapid urbanization within the last year, and the pronounced temporal variation in those ratios appears to reflect this fact. As described by [45], higher levels of urbanization lead to greater MP pollution. However, when the ANOVA was implemented for the effluent, no statistically significant differences were observed for the MPP:MP ratio (*F-test* = 0.857, *p* = 0.373) or FB:MP ratio (*F-test* = 0.814, *p* = 0.385), indicating a similar and stable performance of the sewage plant efficiency during the studied period, despite a higher load of MPPs during 2020.

The microfiber form was also the dominant shape in all seasons, decreasing during the Summer ($0.77 \pm 0.28$ items L$^{-1}$) compared to Autumn ($1.06 \pm 0.28$ items L$^{-1}$), Winter ($1.34 \pm 0.44$ items L$^{-1}$), and Spring ($2.72 \pm 0.90$ items L$^{-1}$), but without statistically significant differences (*F-test* = 2.331, *p* = 0.100). This absence of variations across seasons, together with a similar fiber length by seasons (*F-test* = 0.775, *p* = 0.509) should be associated with the important amounts of synthetic fibers released from washing machines, depending on textile properties, washing conditions, type of detergent and softener, and garment weathering [26,28,46]. Modifications on washing machine filters would be an effective and simple way of preventing FB from entering sewers, as well as using washing bags that act as a microfiber filter between the synthetic clothing and the drain [19,47].

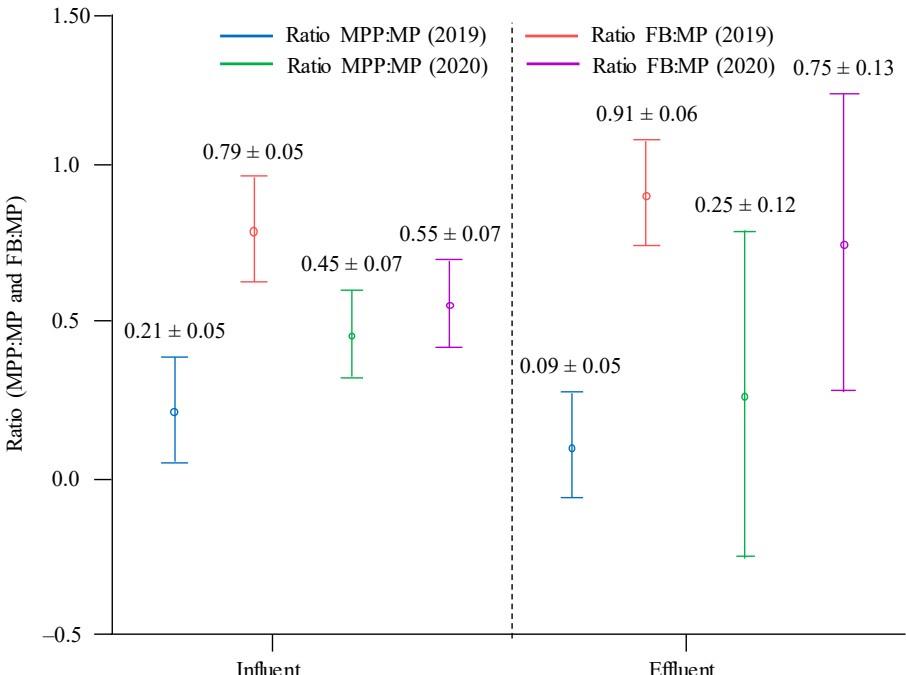

**Figure 3.** Annual variations of MPP:MP and FB:MP ratios (mean ± standard error).

### 3.2. Size, Color, and Polymer Distribution

Figure 4a depicts a total of 10 different colors identified in all samples, with blue being the most common color for fibers (67.39%), while MPPs were mostly white (48.94%) followed by beige (19.15%), brown (8.51%), blue (6.38%), and black (6.38%), similar to that reported in another WWTP with rapid sand filtration [16].

White, blue, and clear colors have proven to be similar to those of plankton, a primary food source for fish [48]. Fibers were by far the most common type in both size groups; i.e., mini-microplastics (<1 mm) and microplastics (1–5 mm), accounting for 88.07% and 72.13%, respectively. The main size of MPs, both in INF and EFF samples, was between 1–2 mm (Figure 4b). The minimum size corresponded to a 210 μm polypropylene film collected in an INF sample, as depicted in Figure 2j, and the maximum dimension corresponded to a white transparent 12 mm fiber also collected in an INF sample. The average size increased from fragments (633.33 ± 79.19 μm) to films (1012.71 ± 235.78 μm) and fibers (1487.11 ± 104.64 μm), which implies that fibers are likely from domestic washing [23].

As previously reported [49], the removal rate of MPs proved to be related to their size, as the removal rate increased with decreasing size: fragments, with the lowest average size, were totally removed in the EFF, and films and fibers were removed at 81.35% and 56.16%, respectively. Several mechanisms have been proposed; i.e., a lower residence time for smaller MP than for larger plastic debris [50], small MP fragment in much higher abundance than large ones [51], and aggregation and settling into the sludge for the lowest sizes [52]. Furthermore, the percentage of MPs smaller than 500 μm increased from the INF (18.63%) to the EFF (21.43%), after UV disinfection, similar to those results reported after chemical disinfection, with a degradation level depending on contact time, temperature, and disinfectant concentration [41,53].

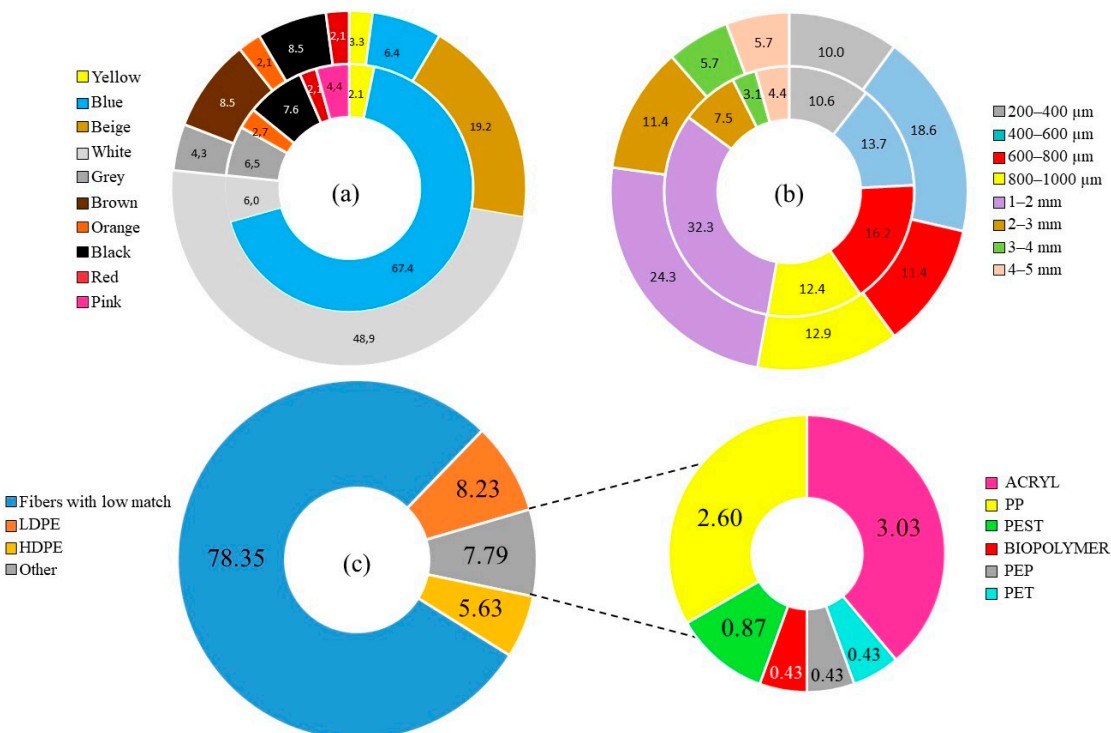

**Figure 4.** Accumulated percentages across the WWTP: (**a**) colors by microplastics shape: Inner ring means FB, outer ring represents MPP; (**b**) size categories based on Spanish Environmental Ministry classification: Outer ring represents EFF; (**c**) polymer types identified by FTIR, including fibers with a low percentage match.

Eight different polymer types were identified across wastewater samples by FTIR, as depicted in Figure 4c. FTIR allows the identification of chemical bonds, giving a characteristic "fingerprint" region [39]. The identification of polymer composition of MP was carried out by means of different reference libraries, containing spectra of all common polymers; i.e., Hummel Polymers and Additives Library (2011 spectra), Polymer Additives and Plasticizers (1799 spectra), Sprouse Scientific Systems Polymers by ATR Library (500 spectra), and Rubber Compounding Materials (350 spectra). The standard criteria reported by [54] were applied, regarding a percentage match over 60% between sample and reference spectrum. In this sense, 78.35% of MPs corresponded to unidentified microfibers that did not match this percentage. Their small size and thickness, as well as the presence of additive compounds, pigments, and dyes, could mask the FTIR signal, together with pollutants closely adhered to their surface [55]. Photo-degradation and weathering are also two factors that could alter the polymer spectra, hindering comparisons with reference libraries [3]. Difficulties in visual classification under a stereomicroscope have been previously reported by [56] in airborne samples, with 62.3% of unidentified fibers because of their small sizes or light colors, or by [36] in wastewater samples, with 63.9% of fibers with a low percentage match or plastic pigments that masked the Raman signal. In this sense, the identification as a synthetic fiber was assessed by practical experience and the criteria previously reported, as no cellular or organic structure, consistent thickness, three-dimensional bending, clear and homogeneously colored, and no taper toward the ends, and different to natural cellulose or cotton fibers, with a ribbon-like morphology under the microscope [39,57]. From the identified fibers, 14.43% corresponded to polyethylene terephthalate (PET) and 7.22% to HDPE. The absorption bands for PET (Figure 5a) at 3100–3400 $cm^{-1}$ correspond to the aromatic C–H stretch, 1730 $cm^{-1}$ identifies the carbonyl group (C=O), 1300–1600 $cm^{-1}$ identifies the aromatic ring, and 1027 $cm^{-1}$ identifies the in-plane C–H stretch [58,59]. Because PET was only found in microfiber forms, a synthetic textile source from domestic washing should be suggested.

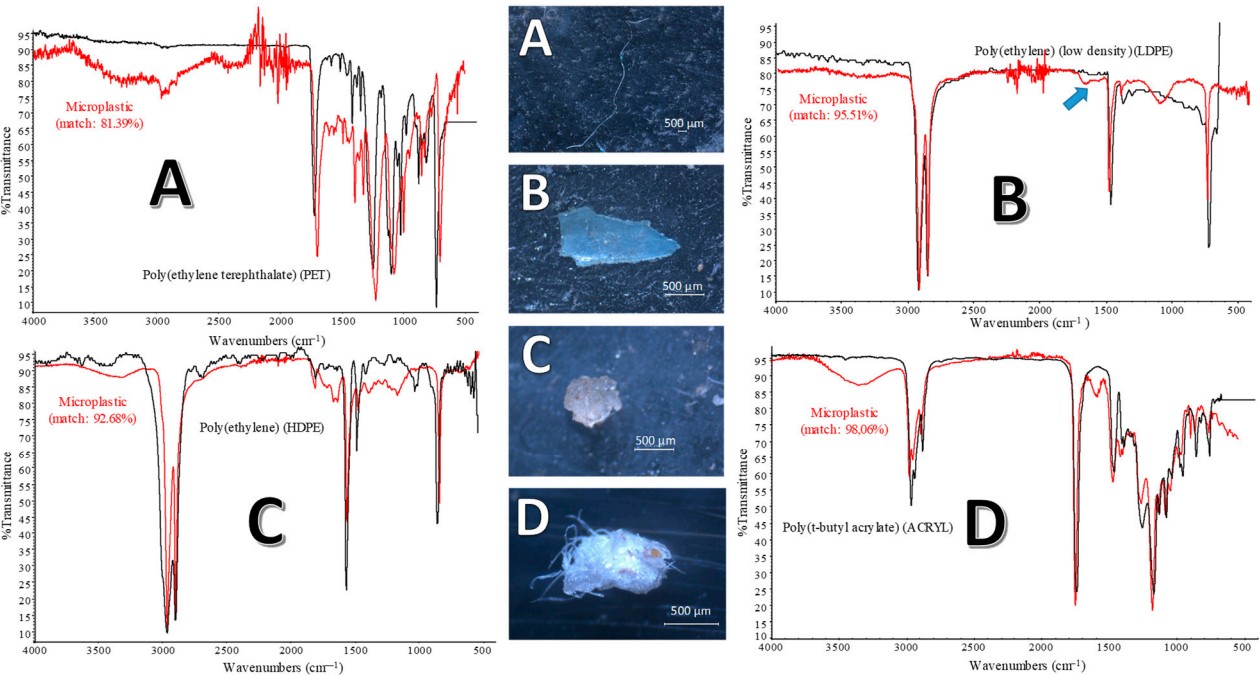

**Figure 5.** Infrared standard spectra (black) and microplastics (red) for: (**A**) Polyethylene terephthalate (PET) EFF 17 October 2019 (81.39%) (Sprouse Scientific Systems Polymers by ATR Library); (**B**) Poly(ethylene) (low density) INF 28 February 2019 (94.51% match) (Sprouse Scientific Systems Polymers by ATR Library); (**C**) Polyethylene INF 23 January 2020 (92.68% match) (Polymer Additives and Plasticizers). (**D**) Poly (t-butyl acrylate) EFF 19 December 2019 (98.06% match) (Sprouse Scientific Systems Polymers by ATR Library).

Figures 5 and 6 depict distinctive absorption peak band examples for eight identified polymers. The wide range of applications and relatively low cost of polyethylene renders its prolific use, being identified as the predominant polymer in many wastewater studies [3,20,60]. Absorption bands for LDPE (Figure 5b) include an asymmetric vibration of the $CH_2$ group between 2930–2850 $cm^{-1}$ wavenumbers, 1450–1470 $cm^{-1}$ for the bending C–C bond between methylene carbons, and a 700–750 $cm^{-1}$ band due to in-plane rocking of $CH_2$ groups. The distinctive peak near 1715 $cm^{-1}$, marked with a blue arrow, would be related to C=O stretching of the non-ionized carboxylic group because of the oxidation of the polymer [61].

It is important to consider the aggressive and severe conditions surrounding microplastics in a wastewater environment, together with a warm and arid climate that will facilitate the oxidation of polymers. The vast majority of film was identified as LDPE polymer (63.16%), probably due to the proximity of the sewage plant to agriculture crops under plastic mulching, a major source of MPs in the terrestrial environment [62]. LDPE is the most inexpensive plastic film and the dominant covering material in the Mediterranean region, with high transmissivity to thermal radiation [63], reaching the WWTP as a MP through atmospheric transportation.

Polypropylene (Figure 6a) has some spectral bands similar to polyethylene, although the absence of peaks between 700 and 750 $cm^{-1}$ clearly distinguishes them [64]. A blue arrow marks a distinctive peak close to 1715 $cm^{-1}$ wavelength due to PP oxidation. After the tertiary treatment, only PP was significantly reduced in the EFF (*F-test* = 5.819, *p* = 0.023), the removal rates for LDPE and HDPE being 98.72% and 86.71%, respectively.

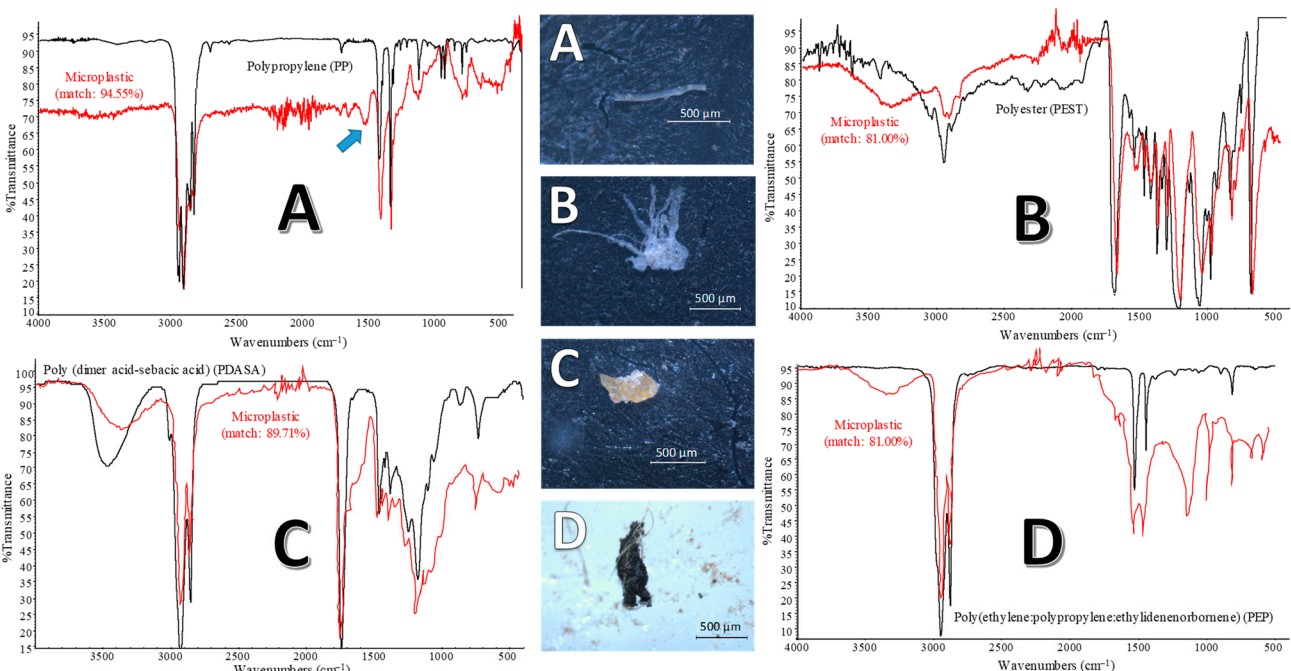

**Figure 6.** Infrared standard spectra (black) and microplastics (red) for: (**A**) Polypropylene (HDPE) INF 24 May 2020 (94.55%) (Polymer Additives and Plasticizers); (**B**) Polyester (PEST) INF 19 September 2019 (81.00%) (Synthetic Fibers by Microscope); (**C**) Poly (dimer acid-sebacic acid) copolymer (PDASA) INF 20 March 2019 (89.71%) (Polymer Additives and Plasticizers); (**D**) Poly (ethylene:propylene:ethylidene norbornene) (PEP) INF 28 February 2019 (81.00%) (Hummel Polymer and Additives).

Acrylate copolymers; i.e., ethylene/ethyl acrylate copolymer and poly (t-butyl acrylate) (Figure 5d) were also identified, as previously reported by [65] in atmospheric microplastics deposited on terrestrial plants, and used in commercial shower gels, peelings, waterproof sunscreen or as a gallant for lipstick [66].

Copolymers as poly (dimer acid-sebacic acid) copolymer (PDASA) (Figure 6c) and poly (ethylene:propylene:ethylidene norbornene) copolymer (PEP) (Figure 6d) were also collected. Studies have reported PDASA biopolymer to be degradable, both in vitro and in vivo, leaving an oily dimer acid residue after hydrolysis [67]. Because of a quick and extensive degradation, this biopolymer completely disappeared in the EFF. PEP was also identified and reported by [68] in WWTPs, mainly used in automobile parts and in the production of pipe seals.

## 4. Conclusions

This paper presented the results of the occurrence, analysis, and removal of microplastics from a full-scale WWTP located in Southeast Spain. An environmentally friendly, reproducible and cheap method for the purification and isolation of microplastics based on density separation was used, avoiding handling with chemicals that may alter their properties and composition. FTIR spectroscopy allowed us to detect 72.41% of microplastics within all microparticles isolated, with average concentrations of $2.74 \pm 0.49$ MP L$^{-1}$ in the influent and $0.98 \pm 0.27$ MP L$^{-1}$ in effluent samples. The removal rate was 64.26%, within the average retention for European WWTPs, contributing to a daily calculated emission of $1.6 \times 10^{7}$ microplastics. The main collected shapes were microfiber (79.65%), film (11.26%), and fragment (9.09%), without microbeads identified in wastewater samples. The dominance of microfiber in all seasons, mainly identified as PET fibers, should be associated with textiles and garments made of synthetic materials shedding fibers during their washing process. Washing at low temperatures, avoiding long washing cycles with washing machines completely full of clothes, and the use of softeners, could decrease friction between garments and reduced the amount of fiber released. Besides, the installation of washing machine filters could avoid their release into sewage plants. The

removal efficiency for particulate forms of microplastics (90.03%) proved to be higher than for microfibers (56.16%), indicating that tertiary treatments are overall more efficient in removing particulate than fiber forms and suggesting an improvement in sand filtration mechanisms by means of coagulants. Differences in annual variations of microplastic burden in the influent, probably due to the proximity to an expanding urban area, were safely cushioned in the effluent, reporting a stable performance of the sewage plant. Lower microplastic size in effluent than influent samples could be related to UV disinfection, and the massive amount of LDPE film identified could be due to the proximity of the sewage plant to greenhouse agricultural crops. Future efforts related to microplastics in WWTPs should be devoted to following four different strategies: (1) a source control, avoiding a massive and indiscriminate use of plastic items, and exploring alternative materials to efficiently reduce marine plastic litter; (2) an effective waste management for the recovery of resources from plastics, preventing their entry into the food chain and water bodies in the form of microplastics; (3) an improvement in international policy legislation, public initiatives, and regulations for the use of plastics; and (4) microplastic-targeted treatment processes in WWTPs to reduce their emission to the environment.

**Supplementary Materials:** The following are available online at https://www.mdpi.com/article/10.3390/w13101339/s1. Table S1. Description of sewage treatment stages and working parameters. Table S2. Physicochemical and biological parameters. Table S3. Samples and volumes collected in "La Aljorra" WWTP for this study. Figure S1. Location of "La Aljorra" WWTP (37°41′16″ N, 01°03′13″ W).

**Author Contributions:** Conceptualization, J.B. and J.L.-C.; Methodology, J.B. and S.O.; Writing, J.B.; Data acquisition, S.O. and J.L.-C.; Visualization, J.B. and S.O.; Funding acquisition, J.L.-C. All authors have read and agreed to the published version of the manuscript.

**Funding:** This work was financed by Project 5245/18IQA (Cetenma and Hidrogea). Analyses carried out by Sonia Olmos were supported by a grant from Fundación Séneca (20268/FPI/17).

**Institutional Review Board Statement:** Not applicable.

**Informed Consent Statement:** Not applicable.

**Data Availability Statement:** The datasets generated and analyzed during the current study are not publicly available due to data belongs to the company operating the sewage plant, but they are available from the corresponding author on reasonable request.

**Acknowledgments:** Authors gratefully acknowledge the work and cooperation of the personnel of "La Aljorra" WWTP with wastewater samples collection.

**Conflicts of Interest:** The authors declare no conflict of interest.

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
