# Peer review of "Assessment of Microplastics in a Municipal Wastewater Treatment Plant with Tertiary Treatment: Removal Efficiencies and Loading per Day into the Environment"

_water, doi:10.3390/w13101339_

Round 1

Reviewer 1 Report

The manuscript deals with very important and interesting topic. It concerns the assessment of microplastics composition in municipal wastewater treatment plant. The content presented in paper are not a new thing but the subject matter is very popular at this time. It means that the impact of this work will be global.

Authors analysed influent and effluent to the wastewater treatment plant according to microplastics using microscope analyse (to confirm shape, size of particles), FTIR analyses to asses type of polymer.

The abstract precisely describe general content of the whole work. The Authors have carefully prepared review of the literature taking into account the latest publications in this research area.

The experiment is transparent, the results are  repeatable and verifiable

The work perfectly fits to the scope of the Journal. It does not require many corrections. The overall aim of the study is stated and completed. It is therefore suggested to accept the article to the publication in the Journal; however, prior to publication some revisions are required.

I recommend this manuscript to the publication in Water Journal. I would like to congratulate the Authors. I enjoyed reading their works.

Suggested correction:

Line 220 – 222: please unified the fonts size

Reviewer 2 Report

Paper talks about microplastics and their removal at WWTP. However, the major concern is whether the samples were collected using the retention time of the WWTP or were they grab samples. If they were grab samples, then this analysis is not true. Analysis needs to be redone and not labeled as removal instead it can be occurence of the samples.

Reviewer 3 Report

The manuscript concerns the important issue of the removal of microplastics from wastewater in an urban wastewater treatment plant located in Southeast Spain, including an oxidation ditch, rapid sand filtration, and ultraviolet disinfection. Eight different polymer families were identified, LDPE film being the most abundant form, with 10 different colors and the main size between 1-2 mm. The following remarks should be taken into account: Manuscript is well prepared, however according to the journal’s guidelines, the Authors should add DOI in the references. Please, add some information about future efforts concerning source control and improvement of legislation for microfiber removal.

Round 2

Reviewer 2 Report

Authors have addressed my comments